# Analysis of Circulating Immune Subsets in Primary Colorectal Cancer

**DOI:** 10.3390/cancers14246105

**Published:** 2022-12-12

**Authors:** Can Lu, Josefine Schardey, Ulrich Wirth, Viktor von Ehrlich-Treuenstätt, Jens Neumann, Clemens Gießen-Jung, Jens Werner, Alexandr V. Bazhin, Florian Kühn

**Affiliations:** 1Department of General, Visceral, and Transplant Surgery, Ludwig-Maximilians-University Munich, 81377 Munich, Germany; 2Institute of Pathology, Medical Faculty, Ludwig-Maximilians-University Munich, 81377 Munich, Germany; 3Department of Medicine III, University Hospital, Ludwig-Maximilians-University Munich, 81377 Munich, Germany; 4German Cancer Consortium (DKTK), Partner Site Munich, 81377 Munich, Germany; 5Bavarian Cancer Research Center (BZKF), 91054 Erlangen, Germany

**Keywords:** colorectal cancer, flow cytometry, immunophenotype, diagnostic model, differentially expressed genes

## Abstract

**Simple Summary:**

The immune system has a vital role in shaping the development and progression of CRC. Circulating immune subsets are the primary resources of tumor-infiltrating immune cells that could directly count against the tumor. The status of the systemic immunity in CRC patients is still unclear. Our study aims to comprehensively evaluate the circulating immune subsets and gene expression profiles of CRC patients. Here, we show that CRC patients have a more prominent systemic immune suppression than healthy controls, as well as NR3C2, CAMK4, and TRAT1, that might involve regulating the number of circulating T helper cells in CRC patients. The distribution of circulating immune subsets in CRC could complement the regional immune status of the tumor microenvironment and contribute to the discovery of immune-related biomarkers, for the diagnosis of CRC.

**Abstract:**

The development and progression of colorectal cancer (CRC) are known to be affected by the interplay between tumor and immune cells. However, the impact of CRC cells on the systemic immunity has yet to be elucidated. We aimed to comprehensively evaluate the circulating immune subsets and transcriptional profiles of CRC patients. In contrast to healthy controls (HCs), CRC patients had a lower percentage of B and T lymphocytes, T helper (Th) cells, non-classical monocytes, dendritic cells, and a higher proportion of polymorphonuclear myeloid-derived suppressor cells, as well as a reduced expression of CD69 on NK cells. Therefore, CRC patients exhibit a more evident systemic immune suppression than HCs. A diagnostic model integrating seven immune subsets was constructed to distinguish CRC patients from HCs with an AUC of 1.000. Moreover, NR3C2, CAMK4, and TRAT1 were identified as candidate genes regulating the number of Th cells in CRC patients. The altered composition of circulating immune cells in CRC could complement the regional immune status of the tumor microenvironment and contribute to the discovery of immune-related biomarkers for the diagnosis of CRC.

## 1. Introduction

Colorectal cancer (CRC) is the third most frequent carcinoma and the second leading cause of cancer-associated death globally, accounting for 1.8 million new cases and 900,000 deaths annually [1]. Despite significant advances in diagnostic and therapeutic options in the past decades, nearly 25% of the patients have synchronous metastases at the initial diagnosis, and virtually 50% of primary CRC patients could develop distant metastases during this disease [2]. Furthermore, after receiving a completed resection of CRC, the 5-year survival rate is approximately 60%, while the rate drops to 12% in the metastatic disease [3]. The therapeutic strategies ought to be improved, in terms of the high rate of metastasis and corresponding worse prognosis in patients with primary CRC. Currently, many biomarkers have been proposed to predict the responses to clinical treatment and to stratify CRC patients according to the risk classification, including miRNAs, circulating tumor cells, circulating tumor DNA, and genetic mutations [4]. However, these approaches are virtually characterized by a tumor-cell-centric nature, overlooking the intrinsic heterogeneity of the tumor microenvironment (TME) and immune elements. The accumulating studies indicated that the immune system has a fundamental role in shaping the development and progression of CRC [5,6,7]. Although most studies focus on tumor-infiltrating lymphocytes (TILs), immune subsets in the peripheral blood are the primary resources for intratumoral immune events. Therefore, the composition and phenotype of circulating immune cell subsets may be linked to the immune response inside the tumor, potentially playing a significant role in predicting the tumor progression and drug responses in CRC. In addition, the impact of CRC on the systemic immunity remains to be elucidated.

Even though acting as the primary effector cell of humoral immunity, B lymphocytes are poorly investigated in the TME because of their controversial role in regulating tumor progression [8]. Conversely, T-cell infiltration of TME has been widely researched in CRC patients [9,10,11,12,13]. Innate immune cells that principally comprise neutrophils, monocytes, dendritic cells (DCs), myeloid-derived suppressor cells (MDSCs), NK, and NKT cells, are also involved in the interplay with tumor cells. Neutrophils are indispensable immune cells to defend against invading microorganisms and facilitate wound healing [14]. Moreover, monocytes are considered critical regulators of cancer development and metastasis, with different subsets performing opposing roles in enabling the tumor growth and preventing the metastatic spread of tumor cells [15]. DCs, one of the essential antigen-presenting cells, could initiate adaptive immune responses and secret the costimulatory molecules to drive the cytotoxic T cells’ clonal expansion [16]. Moreover, MDSCs consist of a heterogeneous population of an early-stage (E-MDSC), monocytic (M-MDSC), and polymorphonuclear origin (PMN-MDSCs) that typically arise in chronic inflammatory sites, including cancer [17]. NK and NKT cells are innate-like lymphocyte populations with cytotoxic functions, independent of the MHC molecules on pathogenic cells and tumor cells in the innate immunity. It is worth noting that the composition of the above immune subsets is seldom reported in the peripheral blood of CRC patients.

This study aimed to comprehensively evaluate the circulating immune subsets and gene expression profiles of CRC patients. Furthermore, peripheral blood immune cell profiles were subsequently used to construct a diagnostic model and correlate with the clinical test data. Our study revealed that CRC patients have a significantly suppressed systemic immunity, compared to healthy controls. 

## 2. Materials and Methods

### 2.1. Study Population

This study included 12 patients with CRC who were diagnosed, according to the 2019 World Health Organization classification, and underwent a curative surgical resection at the Ludwig-Maximilians-University Munich (LMU) hospital (Munich, Germany) between September 2020 and September 2021. Blood samples from these patients were collected within 4 h prior to surgery. The inclusion criteria were a surgical R0 resection, a tumor node metastasis (TNM) stage 0-III, a histologically confirmed colorectal carcinoma, and the provision of written informed consent. The exclusion criteria were a history of chemoradiotherapy treatment, concomitant immune-associated disorders and other carcinomas, and the use of immunomodulating drugs or oral steroids within the past three years. In addition, the pre-operative clinical data of CRC patients were also collected. Peripheral blood samples from 11 healthy donors were obtained from LMU hospital after obtaining written consent, and these samples were considered healthy controls. This study was approved by the local review board.

### 2.2. Flow Cytometry Data Analysis

The procedure of the flow cytometry antibody staining is depicted in the Appendix A. Circulating B lymphocytes, T lymphocytes, and innate immune subsets from whole blood samples were detected using three multicolor flow cytometry panels, respectively (Appendix A). At least 1 × 10^5^ events per sample were acquired promptly after staining by 18-color flow cytometry, using the LSR Fortessa (BD Biosciences) with BD FACSDiva^TM^ software version 8.0.1 (BD Biosciences). For each experiment, the optimal cytometer values were maintained by this software. The flow cytometer setup and performance tracking were conducted using the cytometer setup and tracking beads (BD Biosciences). According to the manufacturer’s protocol, the compensation control was carried out with the CompBeads set (BD Biosciences). The positive staining cells were identified using fluorescence minus one (FMO) control, when necessary [18]. The FMO controls were used for IgM, CD38, CD27, CD10, CD24, IgD, and CD20 in flow panel 1. Eight FMO controls were set for flow panel 2, including CD197, CD194, CD38, CD25, CD196, CD127, CD45RO, and HLA-DR. Furthermore, the FMO controls were separately prepared for CD69, HLA-DR, CD14, CD33, CD16, CD11c, CD15, CD11b, CD66b, and CD56 in flow panel 3. The immunophenotyping of the circulating B lymphocytes, T lymphocytes, and innate immune subsets is shown in Appendix A. In addition, the sequential gating strategy for each panel was depicted in Appendix A, respectively. The expression levels of the immune markers on the circulating B cells, T cells, and innate immune subsets were evaluated by the percentage of the targeted cells or the median fluorescence intensity (MFI) or the absolute number of immune cells. FlowJo software version 10.4 (Tree Star) was applied to analyze the datasets, and the data were displayed in dot plots.

### 2.3. Construction of a Diagnostic Model

The univariable logistic regression was conducted to evaluate the predictive ability of each immune subset, in two cohorts. In order to obtain the immune subsets that displayed a relatively higher accuracy with the prediction, we kept those immune subsets with *p*-values less than 0.05. The support vector machine (SVM) learning model was performed to identify the optimal parameters from the above immune subsets to discriminate CRC from the healthy controls. To ensure the stability and reliability of our prediction method, a tenfold cross-validation was applied by the SVM model. The best parameters were identified from the maximum cross-validation results. The selected parameters were fitted into a multivariable logistic regression analysis to construct the diagnostic model. Each parameter would be assigned a logistic regression coefficient, and an immune score was generated using the following formula: $$Immune\;Score=\overset{\mathrm{Num}}{\underset{n=1}{\sum }}\left(Composition_{n}\times LC_{n}\right),\;$$
where Num refers to the number of immune subsets, *Composition_n_* represents the percentage of the immune subset_n_, and *LC_n_* is the logistic coefficient of the immune subset_n_.

Furthermore, a nomogram was constructed to visualize this diagnostic model in our cohort. The calibration curve and the Hosmer–Lemeshow test were performed to evaluate the goodness-of-fit of the nomogram model. A decision curve analysis (DCA) was used to assess the model’s reliability by calculating the clinical net benefit for patients at each threshold probability. The receiver operating characteristic (ROC) curve was applied to evaluate the discrimination performance of the nomogram. 

A logistic regression analysis was performed using the *stats* R package [19]. A SVM model analysis was conducted using the *e1071* R package [20]. The *pROC* and *ggplot2* R packages were used to draw the diagnostic ROC curves [21,22].

### 2.4. Gene Expression Profile Collection and Processing

The Gene Expression Omnibus (GEO) database was thoroughly searched to find the eligible GEO datasets, based on blood samples with the following searching strategy (“colon” or “colorectal” or “rectal”) and (“cancer*” or “neoplas*” or “dysplasia”) and (“homo sapiens”) and (“gse”). The inclusion criteria of the datasets are listed in Appendix A. A total of two datasets (GSE164191 and GSE46703) representing different independent studies of CRC were enrolled, of which GSE164191 contained 59 CRC and 62 normal samples, and GSE46703 included 14 CRC samples without prior treatment. Moreover, GSE164191 and GSE46703 were derived from GPL570 and GPL6884, respectively. The *GEOquery* R package was used to download the expression matrixes of the above datasets [23]. The probes were annotated into gene symbols, based on the corresponding annotation files. When multiple probes matched one gene, the median was calculated as its expression value. Moreover, since GSE164191 and GSE46703 were hybridized into two distinct platforms, the combat function of the *sva* R package was applied to integrate two normalized datasets into a meta-cohort, to remove the batch effects (Appendix A) [24]. Next, the merging datasets were quantile normalized with the normalizeBetweenArrays function of the *limma* R package (Appendix A) [25]. Therefore, the merged GEO datasets were considered the normalized expression profiles of the blood samples for the CRC and healthy controls. 

Due to the potential interaction between the colorectal tumors and peripheral blood, the sequencing data of the CRC tissue samples were also obtained from the public repository. The Cancer Genome Atlas (TCGA) projects deposited the largest tissue expression matrixes of CRC on the single dataset level. Then the gene expression profiles of 568 CRC patients and 51 non-cancerous samples were downloaded from TCGA through the GDC data portal.

### 2.5. xCell Algorithm

The *xCell* R package was used to deconvolute the peripheral blood mononuclear cell types, based on the merged GEO datasets. By applying a novel gene signature-based method, the xCell algorithm could reliably estimate the enrichment of 64 stromal and immune cell types from the gene expression data derived from tissue or blood samples, among which 34 cell types are immune subsets [26]. According to the validation results of the extensive in-silico simulations and the cytometry immune profiling, xCell outperformed other digital dissection methodologies, including CIBERSORT [26].

### 2.6. Differential Expression Analysis

To identify the differentially expressed genes (DEGs) in the blood and tissue samples between the CRC and normal subjects, we performed the differential expression analysis on the merged GEO and TCGA datasets using the *limma* and *DESeq2* R packages, respectively [25,27]. In the GEO dataset, the DEGs were regarded as any gene with adjusted *p* values of <0.05 and |log2 (Fold change)| > 0.25. Owing to the entity of colorectal carcinoma, the DEGs of the TCGA dataset were defined as genes with adjusted *p* values of <0.05 and |log_2_ (Fold change)| > 1. Furthermore, DEGs that consistently changed in the above two datasets were identified as the common DEGs. 

### 2.7. Gene Ontology Enrichment Analysis

A gene ontology (GO) enrichment analysis was performed to determine the potential biological function of the identified common DEGs, using the *ClusterProfiler* R package [28]. The GO analysis contained three categories: biological process, molecular function, and cellular components. The cutoff criteria of the *p* values of <0.05 and the false discovery rate (FDR) < 0.1 were regarded as statistically significant differences for all analyses.

### 2.8. Correlation Analysis

A correlation analysis was performed to explore the association between the immune cell compositions and genetic expression and to investigate the underlying relationship between the immune cell subsets and clinical test parameters, using the *hmisc* and *corrplot* R packages. The correlation coefficients and corresponding *p* values were used to select the significantly correlated pairs.

### 2.9. Statistical Analysis

The correlation analysis was performed using the Pearson method. The statistical difference between the continuous variables was calculated using the two-sample t-test or the Wilcoxon rank-sum test, depending on the normal distribution. The *p*-values of the multiple testing were corrected using the Benjamini–Hochberg method. The comparisons between the categorical variables were conducted by applying Fisher’s exact test. All statistical analyses were completed using the R software (version 4.1.0). The *p*-value < 0.05 was regarded as statistically significant.

## 3. Results

### 3.1. Patients with CRC Exhibiting a Systemic Immune Suppression

The study was conducted, as depicted in Figure 1. Since chemoradiotherapy could affect the systemic immune system, the blood samples were collected only from patients without neoadjuvant therapy. Table 1 summarizes the clinicopathological characteristics of 12 CRC patients and 11 healthy controls included in the analyses. There was no significant difference in gender between the CRC patients and healthy controls, whereas the CRC patients have a trend towards advanced age, compared to the healthy controls (*p* > 0.05). The detailed data of the immune distributional comparison between these two groups are shown in Appendix A. 

At first, compared to the healthy controls, the proportion of B lymphocytes was significantly (*p* = 0.0421) lower in the CRC patients (Figure 2A,C). There was no distributional difference in the two Breg subsets between the CRC patients and the healthy controls (Figure 2A). No differences in other B lymphocyte subsets were detected between the CRC patients and the healthy controls (Figure 2A). 

Secondly, the distribution of seven subsets belonging to the T lymphocyte population was statistically different in the two cohorts (Figure 2A,C): remarkably lower proportions of circulating T lymphocytes (*p* = 0.0184), and Th cells (*p* = 0.0243), were observed in the CRC patients. In contrast, the percentage of activated CD8T (*p* = 0.0107) and activated Th (*p* = 0.0107) cells was significantly higher in CRC patients compared to healthy controls. Furthermore, CRC patients presented with an increased percentage of the naïve (*p* = 0.0088) and central memory Th (*p* = 0.0088) cells, but with a decreased proportion of the effector Th (*p* = 0.0088) cells, compared to the healthy controls. In addition, the percentage of Tregs and its subsets was comparable between the CRC patients and healthy controls (Figure 2A).

Thirdly, among the four monocyte subsets, only non-classical monocytes were significantly (*p* = 0.0125) lower in CRC patients, compared to the healthy controls. Meanwhile, there was no difference in the percentage of neutrophils between these two groups (Figure 2A). 

Fourthly, similar circulating NK and NKT cell percentages were observed in the CRC patients and healthy controls (Figure 2A). No differences were detected in the CD56^bright^ and CD56^dim^ NK cells (% of NK). Although the CRC patients have a similar distribution of CD69^+^CD56^dim^ and CD69^+^CD56^bright^ NK cells, to the healthy controls, the expression level (MFI) of CD69 on these two NK subsets was significantly (*p* = 0.0277) lower in the CRC patients than in the healthy controls (Appendix A). No differences in other phenotypic markers on the NK and NKT cells were observed between the CRC patients and healthy controls (Appendix A). 

Lastly, in contrast to the healthy controls, the CRC patients also have an increased percentage of the PMN-MDSC (*p* = 0.0107) population (Figure 2C). Moreover, a lower percentage of DCs was detected in the CRC patients than in the healthy controls (Figure 2C).

Furthermore, we applied the digital dissection method of the xCell algorithm to the merged GEO dataset consisting of 73 CRC patients and 62 healthy controls, to estimate the distribution of the circulating immune cells. In total, sixty-four subsets, including thirty-four immune subsets, were calculated for the CRC patients and healthy controls (Figure 2B). Although thirteen immune cell subsets showed a significant difference between the CRC patients and healthy controls (Figure 2D), only the Th cells were consistently changed in the flow cytometry analysis and bioinformatics analysis (Figure 2C,D).

To further characterize the systemic immune status in the different MSI statuses of CRC, we compared the distribution of the circulating fifty-two immune subsets between MSS- and MSI-CRC patients (Appendix A). No differences were detected in the composition of the peripheral immune cells and phenotypic markers expression on the NK and NKT cells between these two cohorts (Appendix A). Therefore, the MSI status did not influence the systemic immunity of the CRC patients. 

### 3.2. Diagnostic Model Allowed for the Differentiation of the CRC Patients from the Healthy Controls

Eleven immune subsets were identified from the univariable logistic regression on fifty-two immune subsets (Appendix A). When the SVM was applied to evaluate the accuracy of the different combinations of the above immune subsets, in discriminating between the CRC individuals and healthy controls, the combination of seven immune subsets was optimal, with an accuracy of 0.936 (Figure 3A), including NC.Monocyte_Leukocytes, E_Th, T_Leukocytes, Activated_Th, Activated_CD8T, Th_Leukocytes, and Naïve_Th (Appendix A). Therefore, these seven immune subsets were chosen to construct a diagnostic model using a logistic regression algorithm. The diagnostic formula was determined as follows:

112.9468 − 44.5381 × Non-Classical Monocyte (% of Leukocytes) − 3.1629 × Effector Th (% of Th cells) − 5.8056 × T (% of Leukocytes) − 6.0697 × Activated Th (% of Th cells) + 8.9753 × Activated CD8T (% of CD8T cells) + 5.7387 × Th (% of Leukocytes) + 0.0072 × Naïve Th (% of Th cells),

Moreover, a nomogram incorporating the above immune subsets was constructed to visualize this diagnostic model and efficiently predict the risk of malignancy (Figure 3B). This study used 12 CRCs and 11 normal samples as the training set. The calibration curve of the nomogram confirmed that the predictive probability of CRC nearly matched the actual probability, which was also supported by the Hosmer–Lemeshow test result (*p* = 1.0000) (Figure 3C). According to the DCA curve, we observed that the diagnostic model acquired the most clinical benefit with the entire range of the threshold probabilities, compared to the individual immune subset (Figure 3D). Furthermore, the ROC analysis in our training cohort suggested that the nomogram model accurately distinguished the CRC and normal subjects with an AUC of 1.000 (95% CI 1.000–1.000), the sensitivity of 1.000, a specificity of 1.000, the positive predictive value of 1.000, and the negative predictive value of 1.000, at the cutoff point of −0.038 (Figure 3E). In addition, each immune cell subset of the model has a good diagnostic performance in distinguishing these two groups with an AUC greater than 0.850 (Figure 3F–H).

### 3.3. NR3C2, CAMK4, and TRAT1 Associated with the Composition of the Th Cells

To further elucidate the underlying molecular mechanism associated with the distinct circulating immune subsets between the CRC cases and healthy controls, we performed the differential expression analysis on the GEO and TCGA datasets. In the GEO dataset, we identified 398 DEGs in the blood samples of CRC patients, compared to the healthy controls, of which 38 genes and 360 genes were up-regulated and down-regulated, respectively (Figure 4A). Meanwhile, 5245 DEGs were obtained from the gene expression analysis on the CRC and non-cancerous tissue samples in the TCGA dataset, including 2594 up-regulated genes and 2651 down-regulated genes (Figure 4B). We performed theintersection of the DEGs between the GEO and TCGA datasets to identify the consistently changed genes in both the blood and tissue samples, regarding the potential interaction between colorectal carcinoma and the systemic immune system. In total, 39 DEGs, consisting of one up-regulated gene and 38 down-regulated genes were identified as the common DEGs (Figure 4C,D). Next, the GO enrichment analysis indicated that these genes were mainly involved in lymphocyte differentiation and purinergic receptor signaling pathway (Figure 4E). 

In order to study the relationship between the different distributional immune subsets and the regulated genes, the correlation analysis between the immune cell subsets and common DEGs was conducted in the healthy controls and CRC patients, respectively (Figure 5A,B). Furthermore, the details of the correlation results are depicted in Appendix A. To ensure the robustness of the selection on the potential genes associated with the composition of the Th cells, the correlation pairs between the immune cell subsets and DEGs were used to filter the genes with a coefficient greater than 0.8 and a *p*-value less than 0.05 in both the healthy controls and the CRC patients. Three genes: NR3C2, CAMK4, and TRAT1, were identified as the candidate genes that may involve the regulation of the composition of circulating Th cells in patients with CRC (Figure 5C–H).

### 3.4. Correlation of the Clinical Test Parameters with the Immune Subsets in the CRC Patients 

To further study the relationship between the fifty-two immune cell subsets and the twelve clinical test parameters, the correlation analysis was performed for the CRC patients using the absolute number of the respective immune subsets from a flow cytometry detection (Figure 6). In order to find the reliable biomarkers associated with the immune cell composition in the peripheral blood, the correlated pairs with the coefficient greater than 0.8 and a *p*-value less than 0.05 were selected from the above analysis. Three parameters were strongly associated with the distribution of the immune subsets in the CRC patients (Table 2). Firstly, the gamma-glutamyltransferase was positively correlated with the level of circulating DCs. Five positively correlated pairs involving the aspartate aminotransferase (AST) were identified, including plasmablasts, activated CD8T cells, effector CD8T cells, class-switched memory B cells, and CD8T cells. In contrast to AST, only two immune subsets, T lymphocytes, and memory Treg cells, were positively associated with the alanine aminotransferase (ALT). 

## 4. Discussion

The immunoscore, based on the quantification of the CD3^+^ and CD8^+^ tumor-infiltrating lymphocytes at the invasive margin and at the core of the carcinoma, has been proven to be more reliable than tumor-node-metastasis (TNM) staging, as a prognostic marker in patients with CRC [29,30]. Meanwhile, cancer immunity is considered a combination of the intratumoral immune system and the systemic immune response [31]. Until now, most publications regarding the systemic immune status of patients with CRC focused on specific cell subtypes and did not consider investigating the majority of the immune cell subsets, simultaneously. To characterize the peripheral blood immune features of the CRC patients, we analyzed fifty-two subsets of circulating immune cells, including B and T lymphocytes, monocytes, neutrophils, NK cells, NKT cells, DCs, and MDSCs. Furthermore, these immune subsets were used to construct the diagnostic model to differentiate the CRC patients from the healthy controls and were further correlated with the clinical test data.

We first demonstrated that the CRC patients have a lower percentage of B lymphocytes than the healthy controls, which is in line with a recent study [32]. However, this finding contradicted Shimabukuro-Vornhagen et al., which showed a comparable proportion of B lymphocytes in the peripheral blood of CRC patients, compared to healthy controls [33]. Since chemoradiotherapy before surgery can potentially change the circulating immune landscape by stimulating the systemic immune response, this discrepancy may be attributed to the different inclusion criteria of patients with CRC. Meanwhile, we observed that multiple subsets of T lymphocytes have different compositional features in CRC patients, than in the healthy controls. T lymphocytes, Th cells, and effector Th cells have a decreased proportion in CRC patients, whereas activated CD8T cells, activated Th cells, naïve Th cells, and central memory Th cells, were significantly increased in those patients. Although evidence reported that the CRC patients have a similar distribution of naïve T cells, central memory T cells, and effector memory T cells with healthy controls [34], they failed to discriminate between the two major subpopulations of T cells, namely CD8T and Th cells. Conflicting results have been reported on circulating Treg cell levels in CRC patients. Dylag-Trojanowska et al. indicated that Treg cells were significantly decreased in CRC patients [35], whereas the opposite trend of Treg cells was reported in another study [34]. Interestingly, our results showed that Treg cells have similar distributional characteristics in CRC patients and healthy controls. In addition, Krijgsman et al. reported no statistical difference in the distribution of T lymphocytes and Th cells, between CRC patients and healthy controls, which is not in line with our study [14]. Due to the critical role of T lymphocytes in the systemic immune reaction, it is fundamental to focus on the dynamic distributional changes of circulating T cells, in the context of leukocytes. Compared to leukocytes as the denominator for T lymphocytes and Th cells in our study, Krijgsman et al. [14] used the lymphocytes or T lymphocytes as the denominator of the above immune cell subsets, partially explaining the discrepant results between the two studies. Due to the low percentage of B and T lymphocytes, and Th cells in the leukocyte population of the peripheral blood, CRC patients have an immune suppression in the adaptive immune response.

To our knowledge, this is the first study comparing circulating DCs of CRC patients and healthy controls. We found that those were significantly less frequent than healthy controls. Furthermore, we demonstrated that CRC patients presented an altered distribution of monocytes, compared to healthy controls, characterized by the reduced proportions of circulating non-classical monocytes. These findings are partially consistent with one clinical study that showed no significant compositional differences in the total monocytes, classical monocytes, intermediate monocytes, and non-classical monocytes, between CRC patients and healthy controls [36]. The explanation for the difference was that in their study CD14^+^CD16^++^ was used to identify non-classical monocytes, whereas we regarded CD14^low/+^CD16^+^ as the immunophenotype of these cells. Regarding DCs and non-classical monocytes belonging to antigen-presenting cells, CRC patients may have an impaired immune activation on the adaptive immune response, due to the low number of these two immune cell subsets in the peripheral blood.

Neutrophils are regarded as critical effector cells in the innate arm of the immune system by counting against the invading microorganisms [37]. Nevertheless, studies on circulating neutrophils are virtually scarce. Our study showed that CRC patients have a similar proportion of circulating neutrophils to healthy controls. Furthermore, it is widely accepted that MDSCs exert immune suppressive effects mostly via inhibiting the T-cell proliferation and stimulating the Treg development [38]. In contrast to several studies that reported that circulating MDSCs were significantly increased in CRC patients [39,40], we found no difference in the distribution of MDSCs between CRC patients and healthy controls. Moreover, our study explicitly indicated that CRC patients had an increased percentage of PMN-MDSCs, compared to healthy controls. Accumulating studies have reported that PMN-MDSCs are the main components of circulating MDSCs and have a more prominent immune suppressive function than M-MDSC [38]. Hence, the high proportions of PMN-MDSCs within the MDSCs population could present a stronger immune suppression on the systemic immune response of CRC patients, than healthy controls.

Additionally, compared to the healthy controls, we demonstrated that CRC patients have an altered phenotype of circulating CD56^dim^ and CD56^bright^ NK cells, characterized by the reduced expression of CD69. Due to CD69 being regarded as the stimulatory membrane receptor of the NK cells [41], both CD56^dim^ and CD56^bright^ NK cells may have a compromised cytotoxic activity in patients with CRC. These findings align with a recent study that showed a reduced expression of activating receptors on the NK cells in CRC patients [34]. Furthermore, Krijgsman et al. proved that the immune suppression of the circulating NK cells could be removed by tumor resection in patients with colon carcinoma [42]. Therefore, CRC could inhibit the immune function of the circulating NK cells via downregulating the expression of the cytotoxic activation receptors.

Furthermore, through bioinformatics analyses on the gene expression profiles of the peripheral blood samples, only the Th cells were consistently identified as the differential immune cells in CRC patients between the flow cytometry detection and xCell algorithm analysis. To explore the underlying molecular mechanism involving the regulation of the Th cells in the peripheral blood, we identified genes differentially expressed, not only in the blood samples, but also in tissue samples of CRC patients, compared to the normal controls, with the consideration of the potential effects of the colorectal tumor on the systemic immune system. Next, we pinpointed three genes that have a strong positive correlation with the level of Th cells in the peripheral blood of both the healthy controls and CRC patients, namely NR3C2, CAMK4, and TRAT1. NR3C2, known as a mineralocorticoid receptor (MR), it has a critical role in mediating a cardiovascular injury induced by the activation of MR. Recent studies revealed that the MR activation could facilitate inflammation by inducing the T lymphocyte differentiation into the pro-inflammatory Th1 and Th17 subsets, while inhibiting the formation of the anti-inflammatory Tregs [43]. CAMK4, a serine/threonine kinase family member, could regulate the gene expression via activating the transcription factors in the cells of immune systems [44]. Previous studies reported that CAMK4, highly expressed in the T cells, was an essential molecule mediating the differentiation of the Th17 cells from the T lymphocytes [45,46]. TRAT1, also referred to as TRIM, can stabilize the T cell receptor (TCR) levels by working as the integral component of TCR [47]. Although lacking studies reported the influence of TRAT1 on the proliferation of T cells, TRAT1 could elevate the expression level of surface CTLA-4 via accelerating its transport from the cytoplasm [48], which may result in the inhibition of the Th cell proliferation. Therefore, NR3C2, CAMK4, and TRAT1, have the potential to be candidate genes involving regulating the number of Th cells in the peripheral blood.

Meanwhile, we established a 7-immune subsets classifier to differentiate the CRC patients from the healthy controls in our cohort. This 7-immune subsets classifier has an excellent performance in diagnosing patients with CRC, according to the corresponding AUC value and the Hosmer–Lemeshow test result. Due to the limited number of CRC patients in our study, we could not conduct the internal validation of our diagnostic model. However, to maximally increase the stability and reliability of this model, we applied the tenfold cross-validation method of the SVM model, to select the best combination of parameters, to build the final diagnostic model. Nevertheless, a large cohort study is needed to validate the diagnostic accuracy of this classifier in the early diagnosis of CRC. 

When investigating the associations between the circulating immune cell subsets and the clinical test parameters, eight positively correlated pairs involving three parameters were identified in CRC patients: AST, ALT, and gamma-glutamyltransferase. Although there were scarcely reports about these relationships in CRC, the above clinical parameters could indirectly reflect the status of the systemic immune profiles, which may contribute to predicting surgery-related complications. 

The primary limitation of this study is that the number of patients is low, which could compromise the validity of our diagnostic model. However, recently, several robust studies constructed the diagnostic model, based on a comparable size of subjects [49,50,51]. In terms of the advanced age in CRC patients, compared to healthy controls, it is difficult to completely eradicate the potential age-related bias in immune cells. Although we failed to validate the expression of NR3C2, CAMK4, and TRAT1 in the Th cells between CRC patients and healthy controls, our study hinted that colorectal carcinoma might affect the expression of these genes, to mediate further the regulation of the circulating Th cells via a direct or indirect interaction. The tumor cell shedding from CRC could directly contact the hematopoietic stem cells, progenitor cells, and circulating lymphocytes, to cause a systemic immunosuppression for the development and progression of the regional CRC, even for metastasis. Moreover, CRC could regulate the systemic immunity via the secretion of soluble biological molecules and extracellular particles. Furthermore, our study characterized the distribution of a broad spectrum of circulating immune subsets and opened new avenues to underlie the molecular mechanisms regulating the composition of the Th cells in the peripheral blood of CRC patients.

## 5. Conclusions

CRC patients displayed profound distinctions in the immune cell subsets’ distribution and their phenotype, compared to the healthy controls, showing that CRC patients have an evident immune suppression in the systemic immune response. Moreover, NR3C2, CAMK4, and TRAT1 were identified as the candidate genes regulating the level of the circulating Th cells in CRC patients, which will be the focus of future studies in our laboratory. These findings are of importance for deciphering the unique features of the circulating immune cell subsets in CRC, which could complement the regional immune status of the TME and contribute to the discovery of immune-related biomarkers for the diagnosis of CRC.

## Figures and Tables

**Figure 1 cancers-14-06105-f001:**
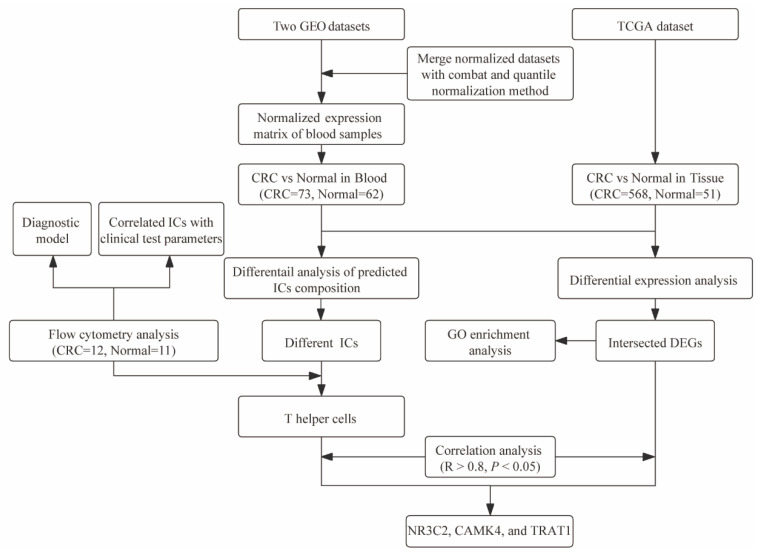
Analysis flow diagram of the study. Abbreviation: GEO, Gene Expression Omnibus; TCGA, The Cancer Genome Atlas; CRC, colorectal cancer; ICs, immune cells; GO, gene ontology; DEGs, differentially expressed genes.

**Figure 2 cancers-14-06105-f002:**
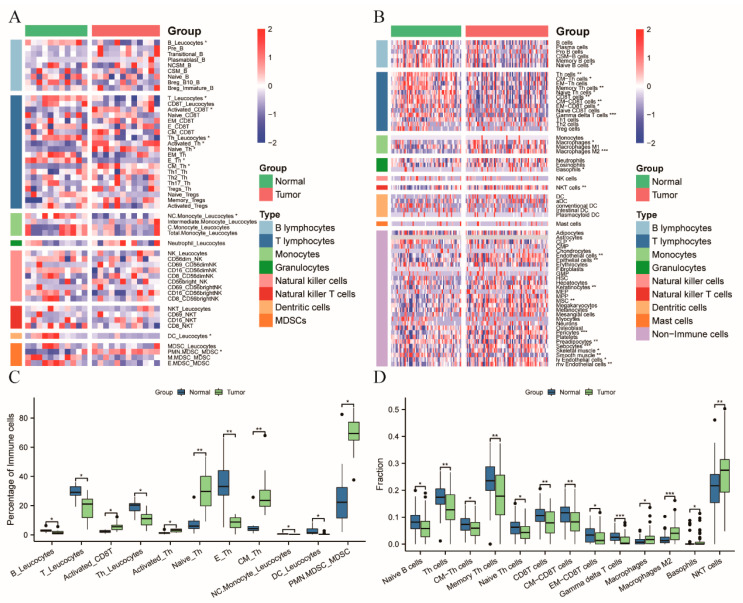
The circulating immune subsets distribution in CRC patients, compared to the healthy controls. (**A**) The heatmap of the immune subsets from a flow cytometry analysis. Each immune subset was expressed as the percentage of the source cells annotated following the underscore; (**B**) The heatmap of the immune and stromal cells computed from the merged GEO datasets using the xCell algorithm; (**C**,**D**) The boxplot of the significantly different immune subsets from the flow cytometry analysis and xCell algorithm, respectively. The bars show the median values of each immune cell subset and the corresponding 95% confidence interval. Corrected *p*-values were calculated for each comparison using the Benjamini–Hochberg method. *, *p* < 0.05; **, *p* < 0.01; ***, *p* < 0.001. Abbreviation: NCSM: non-class switched memory; CSM: class switched memory; Breg, regulatory B cells; EM, effector memory; E, effector; CM, central memory; Tregs, regulatory T cells; NC.Monocyte, non-classical monocytes; C. Monocytes, classical monocytes; NK, natural killer; NKT, natural killer T; DC, dendritic cell; PMN.MDSC, polymorphonuclear MDSC; M.MDSC, mononuclear MDSC; E. MDSC, early-stage MDSC; CLP, common lymphoid progenitors; CMP, common myeloid progenitors; GMP, granulocyte-monocyte progenitor; HSC, hematopoietic stem cell; MEP, megakaryocyte-erythroid progenitor cell; MPP, multipotent progenitor; MSC, mesenchymal stem cell; ly, lymphatic; mv, microvascular.

**Figure 3 cancers-14-06105-f003:**
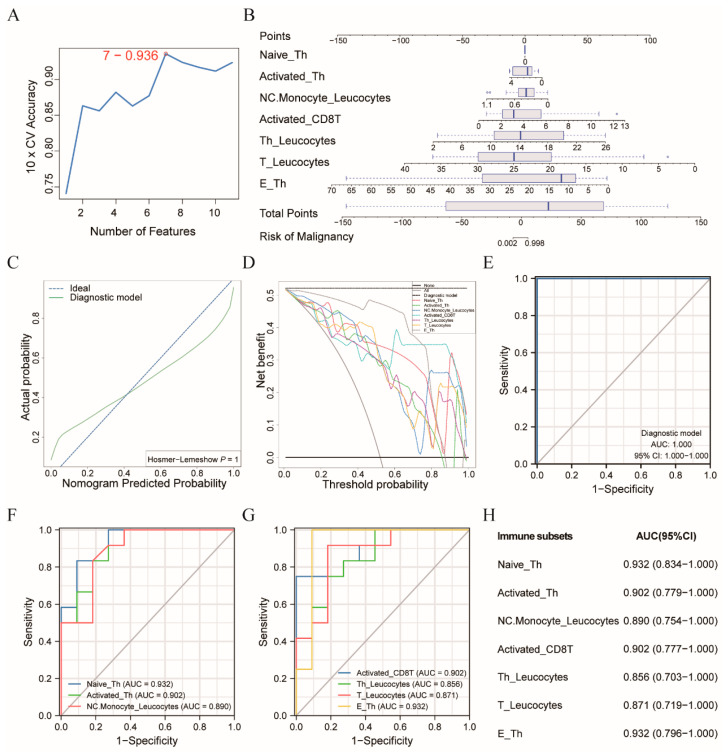
Diagnostic model for differentiating the CRC patients from the healthy controls. (**A**) Tenfold cross-validation accuracy plot of the SVM algorithm; (**B**) Diagnostic nomogram model to predict the risk probability of CRC; (**C**) Calibration curve of the nomogram model; (**D**) DCA curve of the nomogram model and corresponding seven predictive risk factors; (**E**) ROC analysis of the nomogram model; (**F**,**G**) ROC analysis of the seven predictive immune subsets in the model; (**H**) The forest plot of the AUC value and 95% CI for each immune subset. Each immune subset was expressed as the percentage of source cells annotated following the underscore. Abbreviation: Naïve_Th, naïve Th (% of Th); Activated_Th, activated Th (% of Th); NC.Monocyte_Leukocytes, non-classical monocyte (% of Leukocytes); Activated_CD8T, activated CD8T (% of CD8T); Th_Leukocytes, Th (% of Leukocytes); T_Leukocytes, T (% of Leukocytes); E_Th, effector Th (% of Th); Th, T helper, CD8T, CD8^+^ T; ROC, receiver operating curve; AUC, area under the curve; 95% CI, 95% confidence interval.

**Figure 4 cancers-14-06105-f004:**
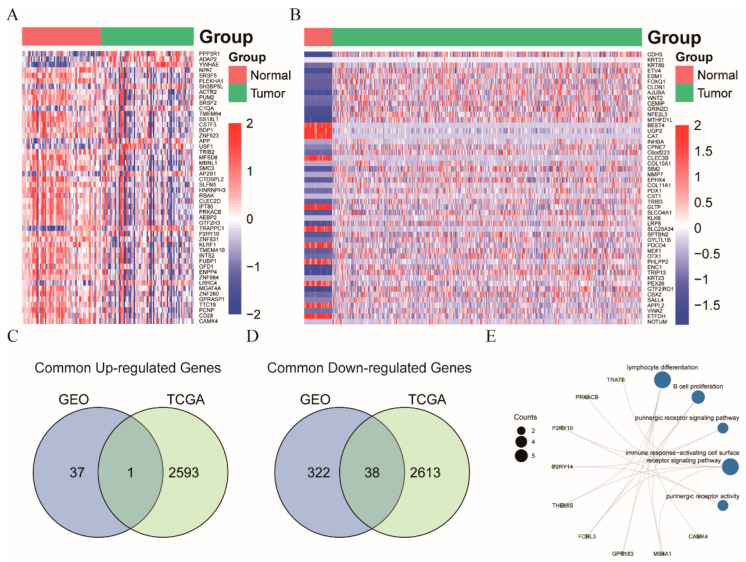
Differentially expressed genes between the normal and CRC in the blood and tissue samples. (**A**) The heatmap of the top 50 DEGs in the GEO dataset; (**B**) The heatmap of the top 50 DEGs in the TCGA dataset; (**C**,**D**) Common up-regulated and down-regulated DEGs between the GEO and TCGA datasets, respectively; (**E**) GO enrichment analysis of the common DEGs. Abbreviation: GEO, Gene Expression Omnibus; TCGA, The Cancer Genome Atlas. DEG, differentially expressed genes.

**Figure 5 cancers-14-06105-f005:**
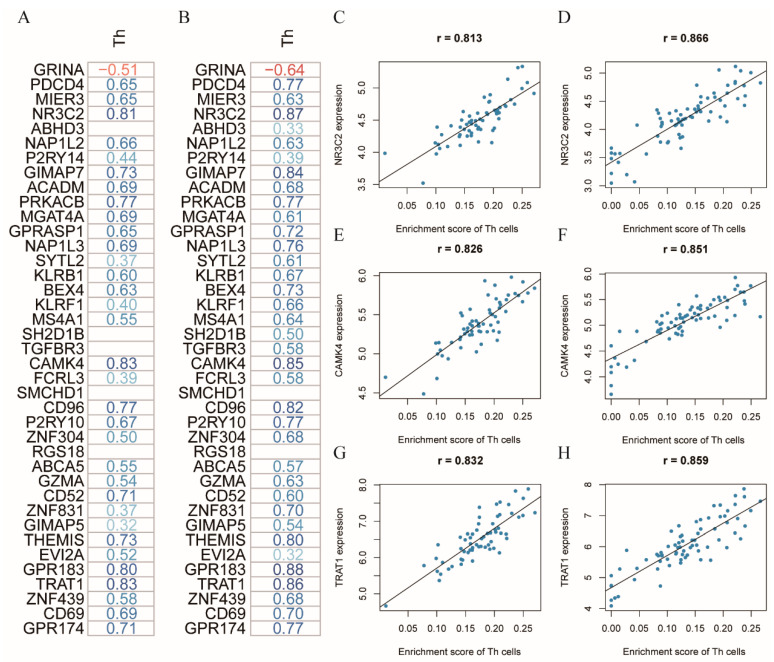
Identification of the candidate genes associated with the composition of the circulating Th cells. (**A**,**B**) The heatmap of the correlation coefficient between the common DEGs and the Th cells in the healthy controls and CRC patients, respectively. Blank cells represented the *p*-value of the correlation greater than 0.05. Blue color and red color referred to the positive and negative correlations, respectively; (**C**,**D**) Correlation plot of NR3C2 with the Th cells in the healthy controls and CRC patients, respectively; (**E**,**F**) Correlation plot of CAMK4 with the Th cells in the healthy controls and CRC patients, respectively; (**G**,**H**) Correlation plot of TRAT1 with the Th cells in the healthy controls and CRC patients, respectively. Abbreviation: Th, T helper.

**Figure 6 cancers-14-06105-f006:**
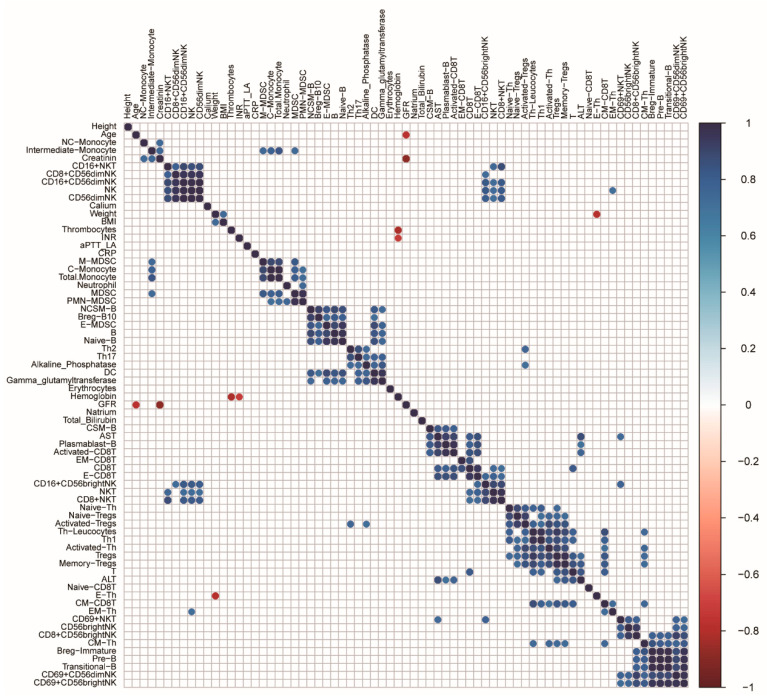
Correlation analysis between the circulating immune subsets and the clinical test parameters. Each immune subset was expressed as the absolute cell number in 200 uL of peripheral blood. Blank cells represented the *p*-value of the correlation greater than 0.05. Blue circles and red circles referred to positive and negative correlations, respectively. Abbreviation: NCSM: non-class switched memory; CSM: class switched memory; Breg, regulatory B cells; EM, effector memory; E, effector; CM, central memory; Tregs, regulatory T cells; NC.Monocyte, non-classical monocytes; C.Monocytes, classical monocytes; NK, natural killer; NKT, natural killer T; DC, dendritic cell; PMN.MDSC, polymorphonuclear MDSC; M.MDSC, mononuclear MDSC; E-MDSC, early-stage MDSC.

**Table 1 cancers-14-06105-t001:** Clinical characteristics of healthy controls and CRC patients.

Variables	CRC (*n* = 12)	Healthy Control (*n* = 11)	*p*-Value
Age, year *	75.0 (69.0, 78.0)	58.0 (53.5, 68.0)	0.0600 ^a^
Gender			0.6800 ^b^
Female	6 (50.0%)	4 (36.4%)	
Male	6 (50.0%)	7 (63.6%)	
Sidedness			
Left side	6 (50.0%)		
Right side	6 (50.0%)		
Elective surgery			
Yes	12 (100.0%)		
Surgery Type			
Open surgery	9 (75.0%)		
Laparoscopic surgery	2 (16.7%)		
Robot-assisted surgery	1 (8.3%)		
T (AJCC 7th)			
T1	1 (8.3%)		
T2	6 (50.0%)		
T3	3 (25.0%)		
T4a	2 (16.7%)		
N (AJCC 7th)			
N0	11 (91.7%)		
N1b	1 (8.3%)		
M (AJCC 7th)			
M0	12 (100.0%)		
Tumor stage (AJCC 7th)			
I	7 (58.4%)		
II	4 (33.3%)		
III	1 (8.3%)		
Residual tumor classification			
R0	12 (100.0%)		
MSI			
No ^c^	8 (66.7%)		
Yes ^d^	4 (33.3%)		
Bethesda			
No	12 (100.0%)		

Abbreviations: CRC, colorectal cancer; AJCC, American Joint Committee on Cancer; T, tumor; N, lymph node; M, metastasis; MSI, microsatellite instability. Data were represented as n (%) unless otherwise annotated. * Age was presented as the median and confidence interval. ^a^ represented the Wilcoxon rank-sum test. ^b^ denoted Fisher’s exact test. ^c^ indicated no protein loss, and ^d^ suggested the loss of expression of MLH1 and PMS2 in the immunohistochemistry.

**Table 2 cancers-14-06105-t002:** Clinical test parameters correlated with the immune subsets.

Clinical Parameters	Immune Cells	Coefficient	*p*-Value
Gamma-glutamyltransferase	Dendritic cells	0.96	1.31 × 10^−6^
AST	Plasmablasts	0.95	1.55 × 10^−6^
AST	Activated CD8T cells	0.91	4.91 × 10^−5^
AST	Effector CD8T cells	0.87	2.60 × 10^−4^
ALT	T lymphocytes	0.84	6.60 × 10^−4^
AST	CSM-B cells	0.83	7.29 × 10^−4^
ALT	Memory Treg cells	0.82	1.19 × 10^−3^
AST	CD8T cells	0.81	1.34 × 10^−3^

Abbreviation: AST, aspartate aminotransferase; ALT, alanine aminotransferase; CSM, class-switched memory; Treg, regulatory T; CD8T, CD8^+^ T.

## Data Availability

The original flow cytometry data can be provided by the corresponding author. The public datasets presented in this study are openly available in the GEO (https://www.ncbi.nlm.nih.gov/geo/, accessed on 15 January 2022) and TCGA (https://portal.gdc.cancer.gov/, accessed on 25 January 2022) databases.

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
