# Peer review of "Analysis of Circulating Immune Subsets in Primary Colorectal Cancer"

_cancers, 2022, doi:10.3390/cancers14246105_

Round 1

Reviewer 1 Report

The manuscript  by Lu et al entitled analysis of circulating immune subsets in primary colorectal cancer describes a distinct distribution of immune cells between patients with CRC and healthy controls and shows that patients with CRC show immune suppression, and they suggest that immune-related biomarkers can support in the diagnosis of CRC.

The manuscript is clearly written and results are presented in a solid way. Despite the interesting findings and novelty in the field, there are some concerns that need to be addressed prior to publication.

1.     As the authors already mention as limitation, the number of patients included in the study is low which can have impact on the validity of the diagnostic model. The authors use all the 12 CRC samples and 11 normal samples as training set. A test set, testing the actual model, is lacking. Perhaps an additional available set can be used to validate and test the predictive value of the model?

2.     Among the samples, a relatively high percentage of MSI tumors are included (33%) This is higher than what is expected from the general population and these tumors are very different compared to MSS tumors in their genetic composition and immunogenicity. The authors should comment/study their differences and how this might have influenced the results.

3.     The patients included had stage I-III tumors. How do the findings on immune suppression relate to patient outcome?

4.     The authors studied the relation between immune cells and laboratory parameters such as AST, ALT and GGT. The association is described, however the authors do not mention whether they are associated with higher or lower levels and whether these levels were increased. In stage I-III CRC, the levels of these liver values are seldom increased. Additionally, the rationale for the correlation of immune cells and these laboratory parameters is lacking.

5.     The authors describe the use of the classifier in early diagnosis of CRC. From the manuscript it is rather unclear what the additional clinical value is in this early diagnosis since the patients used in the study were already diagnosed with CRC. Can they support the additional value by comparing the immune status with patients with premalignant lesions or identify patients at high risk of developing CRC to offer non-invasive tests or intensified screening? How do the authors see this additional value?

Reviewer 2 Report

This manuscript by Lu et al. addresses the role of systemic immunity in the development of CRC. To do this, they take a very detailed look at and compare the subsets of circulating immune cells in CRC patients and healthy subjects. In addition, they perform bioinformatics analyses of selected GEO datasets on circulating blood cells from CRC patients and healthy subjects and TCGA datasets on tissue from CRC patients to complement their findings and gain insight into the characteristics of the observed differences leading to the identification of some genes that could be involved in regulation of subsets of immune cells.

The main result shown in the manuscript is that CRC patients appear to be immunosuppressed to a certain extent, which could be related to the ease of cancer progression and be taken into account to develop diagnostic tools. Therefore, these results could be of great value to improve diagnostic tools for CRC patients and potentially pinpoint novel therapeutical targets in this disease.

The article is well written and correctly structured, although a spell check would be necessary as there are some typographical errors. I consider that this work deserves its publication, but I would like to make some comments that could be addressed by the authors in order to clarify some points and increase the interest of potential readers:

  • One of the possible weaknesses of the study lies in the difference in age observed between CRC patients and healthy controls, since the influence of age on the immunological status of people is well known. There is an age difference between the CRC groups and healthy controls, although it is not statistically significant according to the comparative test carried out by the authors (Wilkonson rank sum test). The application of this non-parametric test suggests that the age data distributions in both groups do not conform to normality. In that case it would be more informative to present medians and confidence intervals as summary values instead of means and standard deviations. In addition, the conclusions of the present study should be extrapolated with caution to patients younger than those included in it.

  • Construction of the diagnostic model is based on a very limited sample of cases and controls. Although the identification of relevant cell subsets using SVM is very accurate and the use of these as variables in a logistic regression model provides very high sensitivity and specificity, this analysis was performed on the same training data set. Wouldn't this be expected? A validation of the proposed diagnostic model on an independent cohort would be more appropriate to show its true usefulness.

  • The bioinformatic analysis work carried out on GEO datasets and TCGA datasets is very interesting and partially supports the conclusions obtained from the analysis of immune cell subsets by flow cytometry. However, it should be taken into account that the establishment of direct comparisons or correlations between both types of evidence can be delicate. In fact, the data from GEO datasets and TCGA datasets would correspond to different cohorts of subjects that may not be homogeneous in their characteristics (age, tumor stages, ...) with the patients and controls analyzed by flow cytometry.

  • Finally, the main idea of the article would be that CRC tumors could modify the behavior of immune cell populations in such a way that their maintenance and progression would be easier. In this sense, the detection of some genes that could be related to this activity would present a link to a mechanistic explanation of what was observed. Although speculative, could the authors comment on possible ways in which tumor cells could affect the expression of these genes in cells of the immune system?

Round 2

Reviewer 1 Report

All raised points have either been added to the manuscript or adequately addressed. The manuscript is suitable for publication in cancers to my opinion.